# Prevalence of Hypovitaminosis C and its Relationship with Frailty in Older Hospitalised Patients: A Cross-Sectional Study

**DOI:** 10.3390/nu13062117

**Published:** 2021-06-20

**Authors:** Yogesh Sharma, Alexandra Popescu, Chris Horwood, Paul Hakendorf, Campbell Thompson

**Affiliations:** 1College of Medicine & Public Health, Flinders University, Adelaide 5042, Australia; 2Department of General Medicine, Division of Medicine, Cardiac & Critical Care, Flinders Medical Centre, Adelaide 5042, Australia; 3Department of Geriatrics & Rehabilitation, Flinders Medical Centre, Adelaide 5042, Australia; alexandra.popescu@sa.gov.au; 4Department of Clinical Epidemiology, Flinders Medical Centre, Adelaide 5042, Australia; chris.horwood@sa.gov.au (C.H.); paul.hakendorf@sa.gov.au (P.H.); 5Discipline of Medicine, The University of Adelaide, Adelaide 5005, Australia; campbell.thompson@adelaide.edu.au

**Keywords:** frailty, vitamin C deficiency, elderly, hospitalisation

## Abstract

Frailty is common in older hospitalised patients and may be associated with micronutrient malnutrition. Only limited studies have explored the relationship between frailty and vitamin C deficiency. This study investigated the prevalence of vitamin C deficiency and its association with frailty severity in patients ≥75 years admitted under a geriatric unit. Patients (*n* = 160) with a mean age of 84.4 ± 6.4 years were recruited and underwent frailty assessment by use of the Edmonton Frail Scale (EFS). Patients with an EFS score <10 were classified as non-frail/vulnerable/mildly frail and those with ≥10 as moderate–severely frail. Patients with vitamin C levels between 11–28 μmol/L were classified as vitamin C depleted while those with levels <11 μmol/L were classified as vitamin C deficient. A multivariate logistic regression model determined the relationship between vitamin C deficiency and frailty severity after adjustment for various co-variates. Fifty-seven (35.6%) patients were vitamin C depleted, while 42 (26.3%) had vitamin C deficiency. Vitamin C levels were significantly lower among patients who were moderate–severely frail when compared to those who were non-frail/vulnerable/mildly frail (*p* < 0.05). After adjusted analysis, vitamin C deficiency was 4.3-fold more likely to be associated with moderate–severe frailty (aOR 4.30, 95% CI 1.33-13.86, *p =* 0.015). Vitamin C deficiency is common and is associated with a greater severity of frailty in older hospitalised patients.

## 1. Introduction

Worldwide, with an aging population, there is a global interest in aging processes and age-related diseases. Frailty represents a precise measurement of aging-related symptoms and constitutes a syndrome taking into account physical disability, low energy levels and loss of cognition [1]. Frailty is associated with poor clinical outcomes such as falls, poor health related quality of life (HRQoL), nursing home placement and death [1,2]. This syndrome has been shown to be potentially preventable and can be reversed if targeted in earlier stages [3]. The prevalence of frailty in acutely hospitalised older patients can be up to 48% [4] while as many as 84% of patients admitted to a geriatric evaluation and management (GEM) unit may be frail [3]. 

While protein energy malnutrition is a major risk factor for the development of frailty, the role of micronutrient deficiencies in the development of frailty is less clear [5]. Micronutrient deficiencies are far more common and often precede development of overt malnutrition [6]. Micronutrient deficiencies can potentially increase the risk of frailty through multiple mechanisms such as an increase in oxidative stress and inflammation, the impairment of bone and muscle metabolism and a reduction in immunity [7]. Vitamin C is a powerful antioxidant and 40% of the total body pool is present in the skeletal muscles [8]. 

Previous studies suggest that micronutrient deficiency (especially vitamin D and vitamin B12 deficiency) may be associated with frailty, but the relationship between biochemical vitamin C deficiency and frailty is unclear [7]. To date, only limited studies [9,10] have explored the relationship between frailty and vitamin C deficiency. The aims of the current research were to determine the prevalence of vitamin C deficiency in older frail patients admitted to a GEM unit and explore the relationship between vitamin C deficiency and severity of frailty. The hypothesis for this research was that older hospitalised patients will have a high prevalence of vitamin C deficiency and low vitamin C levels will be associated with a greater severity of frailty.

Outcomes: The primary outcome for this study was to determine the prevalence of biochemical vitamin C deficiency in older patients admitted to a GEM unit, and the secondary outcome was to determine whether biochemical vitamin C deficiency is a predictor of severe frailty.

## 2. Materials and Methods

Patients ≥ 75 years who were admitted to the GEM unit of Flinders Medical Centre between May–December 2020 were recruited by convenience sampling in this research. A written informed consent was obtained from the participants, and in the case of cognitive impairment, consent was obtained from the legal guardian. A member of the research team approached the participants and provided them with a participant information sheet in addition to verbal information about the research project. The participants were given sufficient time to read and discuss their participation with their caretakers as well as the treating medical team. If the participants were agreeable, they were asked to sign a consent form. 

The exclusion criteria were a lack of a valid consent, patients receiving end of life care and those on vitamin C replacement. Ethical approval for this study was granted by the Southern Adelaide Human Clinical Research Ethics Committee, and this study was registered with the Australia and New Zealand Clinical Trial Registry.

Frailty assessment was performed by use of the Edmonton Frail Scale (EFS). The EFS is a valid and reliable instrument for the identification of frailty in hospitalised patients and predicts clinical outcomes [11,12]. The EFS contains nine components and is scored out of 17. Individual components include: cognition, general health status, self-reported health, functional independence, social support, polypharmacy, mood, continence and functional performance. The component scores are summed, and the following cut-off scores are used to classify the severity of frailty: not frail (0–5), apparently vulnerable (6–7), mild frailty (8–9), moderate frailty (10–11) and severe frailty (12–17). Comorbidities were assessed by use of the Charlson comorbidity index (CCI), which is a score based on various diseases such as myocardial infarction, congestive heart failure, diabetes, renal failure, cerebrovascular disease, peripheral vascular disease, chronic lung disease, liver disease, peptic ulcer and acquired immunodeficiency syndrome (AIDS) and is a valid and reliable method of measuring comorbidity [13,14].

The physical functioning was assessed by use of the Short Physical Performance Battery (SPPB) test, which is a validated measure of lower limb function in older adults and uses tasks that mimics activities of daily living [15]. This test comprises of three subtests: standing balance, four-metre gait speed (4 m GS) and five sit-to-stand (5 STS) tests. The subtests are scored from 0–4 and summated to give a total SPPB score (ranging from 0–12) with higher scores being indicative of a better physical performance [16]. Fall risk was assessed using the Timed Up and Go (TUG) test [17]. In this test, the patient is asked to rise from a seated position, walk 3 metres, turn around and return and sit in the starting point chair while timed. Patients who are unable to complete this test in < 12 seconds are considered to have at high risk of falls [18].

Cognitive status was determined by the use of the Mini Mental State Examination (MMSE) [19] and mood was assessed by using the Geriatric Depression Scale (GDS) [20]. The GDS is a 15-item tool that has been validated for screening depressive symptoms in the older population, including acutely hospitalised medical patients [21,22]. Nutrition risk was determined by use of the Malnutrition Universal Screening Tool (MUST) [23] and the HRQoL was determined by the European Quality of Life 5-Dimension 5-Level (EQ5D-5L) questionnaire [24]. The activities of daily living (ADL) were assessed by use of the Hospital Admission Risk Profile (HARP) score [25], which predicts patients at high risk of discharge to a facility.

Fasting venous blood samples were drawn by a trained phlebotomist. The sample for vitamin C level was wrapped in an aluminium foil and immediately placed on ice for transport to a central laboratory. High performance liquid chromatography (HPLC) was used to determine vitamin C levels. HPLC has been previously validated for the rapid and specific measurement of vitamin C [26]. Plasma vitamin C levels correlate with dietary vitamin C intake and unlike leucocyte vitamin C levels, plasma vitamin C levels are not influenced by changes in the white blood cell (WBC) count and thus represent an accurate measure of vitamin C status [26,27]. According to Johnston’s criteria [28], vitamin C levels ≥ 28 μmol/L are classified as normal, 11–27 μmol/L as vitamin C depletion and < 11 μmol/L as vitamin C deficiency. For this study, patients with vitamin C levels ≥ 28 μmol/L were defined as vitamin C replete and all those with levels < 28 μmol/L as hypovitaminosis C. Vitamin C levels were also divided into quintiles, and we compared clinical outcomes such as length of hospital stay (LOS), in hospital mortality and readmissions within 30 days of hospital discharge in different quintiles. In addition, the phlebotomist collected venous blood samples for determination of haemoglobin, creatinine, C-reactive protein (CRP), albumin, vitamin D and vitamin B12 levels. The technique of spectrophotometry was used to determine haemoglobin, creatinine and albumin levels, while rapid immunoassays, Roche Diagnostics (www.roche.com), determined the C-RP, vitamin D and vitamin B12 levels in the central laboratory. 

### Statistics

The normality of the data was assessed by visual inspection of histograms. Continuous variables were assessed by use of the Student’s t-tests or rank sum tests and categorical variables by Chi squared statistics or Fisher’s exact test as appropriate. A Kruskal–Wallis H test was used to compare the LOS in different quintiles of vitamin C. Patients with EFS scores ≥10 were classified as moderately to severely frail while those with EFS scores <10 as non-frail, vulnerable or mildly frail. We correlated vitamin C levels with EFS scores. A logistic regression analysis was used to determine whether vitamin C deficiency was associated with a greater severity of frailty after adjustment for the following co-variates: age, sex, Charlson index, smoking status, MUST score, MMSE, GDS, fruit and vegetable intake, socioeconomic status (determined by annual income), HARP score, vitamin D and vitamin B12 levels. We determined any effect modification by use of interaction terms with vitamin D and vitamin B12 in the model. We performed a sensitivity analysis and determined the bias-corrected estimates using the jackknife resampling method as suggested by Nemes et al [29]. Prediction graphs with 95% confidence intervals were plotted to determine whether vitamin C levels were associated with increasing severity of frailty using the marginsplot command in STATA. Polypharmacy was defined as being on five or more medications.

The sample size was based on a recent study [30], which suggested a high prevalence (>70%) of biochemical vitamin C deficiency in hospitalised patients. Assuming a prevalence of 70% and a precision level of 10%, the calculated sample size for this study was 140, and assuming 10% missing data, a total of 160 patients were deemed sufficient for this study. All statistical analyses were conducted using Stata version 17.0 (StataCorp, College Station, TX, USA).

## 3. Results

Six hundred and three patients were admitted under the GEM unit between May and December 2020, of whom, 176 patients were approached by convenience sampling for participation, and 160 patients were recruited for this study while 16 patients were excluded because of various reasons (Figure 1).

There was no missing data and vitamin C results were available for all the participants. The mean (SD) age was 84.4 (6.4) years (range: 75–105 years) and 96 (60%) were females. All patients were residing in their own homes and 78 (48.7%) were living with their partners. The mean Charlson index was 8.4 (2.6), and the majority of patients were on polypharmacy 130 (81.3%) and many were admitted with falls as the principal diagnosis (69, 43.1%). The mean (SD) vitamin C levels were 26.8 (23.0) μmol/L, (range: 3–148). The median (IQR) time from hospital admission to the collection of vitamin C sample was 4 (4, 4) days. According to Johnston’s criteria, 61 (38.1%) had a normal vitamin C status, 57 (35.6%) were vitamin C depleted and while 42 (26.3%) had vitamin C deficiency (Figure 1). Overall, 99 (61.9 %) were classified as having hypovitaminosis C, i.e., vitamin C levels below 28 μmol/L. The mean (SD) EFS score was 9.8 (2.1) (range: 5–16). Sixty-eight (42.5%) patients were classified as non-frail/vulnerable or with mild frailty (EFS < 10) and 92 (57.5%) were classified with moderate to severely frailty (EFS ≥ 10). Patients who had moderate to severe frailty were more likely to be older, with a higher Charlson index, a lower mean MMSE score and were more likely to be depressed than those who were non-frail/vulnerable or had mild frailty (Table 1). Patients with moderate to severe frailty had poor physical functioning, as reflected by a longer TUG test and lower SPPB scores, and had a poorer health related quality of life, as reflected by the lower EQ5D index when compared to those who were non-frail/vulnerable or had mildly frailty (Table 1). Both LOS and in-hospital mortality was significantly higher among patients who were moderately to severely frail in comparison with the non-frail/vulnerable/mildly frail group. However, there was no difference in 30-day readmissions between the two groups.

There was no difference in the median (IQR) time in days from hospital admission to the collection of the vitamin C sample between the moderate to severely frail and the non-frail/vulnerable/mildly frail groups (4 (3,4) vs. 4 (4, 4) days, *p* > 0.05, respectively). The mean (SD) vitamin C levels were significantly lower among patients who had moderate to severe frailty when compared to those who were non-frail/vulnerable or had mild frailty (22.9 (21.4) vs. 31.8 (24.4), *p* = value = 0.015). There was a weak negative correlation between vitamin C levels and EFS score (correlation coefficient = - 0.14, *p* = 0.081). The logistic regression analysis suggested that patients with vitamin C deficiency were more likely to be associated with moderate to severe frailty after adjustment for age, sex, Charlson index, smoking status, MUST score, MMSE, GDS, fruit and vegetable intake, socioeconomic status, HARP score, vitamin D and vitamin B12 levels when compared to those who were non-frail/vulnerable or mildly frail (aOR 4.30, 95% CI 1.33–13.86, *p =*= 0.015) (Table 2). The senstivity analysis using the jackknife resampling method confirmed these estimates (aOR 4.30, SE 3.01, 95% CI 1.07–17.12, *p =* 0.039). There was no effect modification (*p* > 0.05) by use of interaction terms with vitamin B12 and vitamin D. The margins plot indicated that low plasma vitamin C levels predicted moderate to severe frailty (Figure 2). Clinical outcomes such as LOS, inhospital mortality and 30-day readmissions were not significantly different according to different vitamin C quintiles (> 0.05).

## 4. Discussion

The results of this study suggest that there is a high prevalence of hypovitaminosis C in older hospitalised patients with a quarter of patients being vitamin C deficient. Patients with moderate to severe frailty demonstrated a significantly lower vitamin C levels when compared to those who were non-frail/vulnerable or were mildly frail even after adjustment for various co-variates.

The prevalence of hypovitaminosis C in this study was 61.9%, which confirms the findings of a recent Australian study [30] that also found that a high proportion (>70%) of acutely hospitalised older patients were vitamin C depleted. Similary, a study [31] on patients who were referred for an elective surgical procedure found that 43.1% had vitamin C depletion. The slightly lower prevalence of vitamin C deficiency in the above study could be related to the lower mean age (62 (15.3) years vs. 84.4 (6.4) years) of patients in this study because two previous studies [32,33] have suggested that vitamin C levels decline with aging. 

Previous studies indicate that older people have inadequate intake of micronutrients such as vitamin A, D, E, B6, B12, folate and zinc [34,35,36]. The intake of vitamin C and carotenoids may also decline with older age [37,38]. Older people are at a high risk of micronutrient malnutrition due to a range of factors including anorexia of ageing and social factors, such as difficulty with shopping or the preparation of meals due to physical disabilities [32,39,40]. Our study also found that the risk of malnutrition, as determined by the MUST score, was not significantly higher among patients who had hypovitaminosis C. These findings corroborrate previous evidence that micronutrient deficiencies in older patients may exist even without overt signs of clinical malnutrition [7]. 

This study indicates that vitamin C deficiency is associated with a greater severity of frailty in older hospitalised patients. This could be related to a higher degree of oxidative stress in ascorbic acid deficient skeletal muscles leading to muscle dysfunction. Previous evidence indicates that there is a relationship between oxidative stress and frailty [41,42]. Reactive oxygen species (ROS) are continuously produced in skeletal muscles [43], and the production of ROS is increased by exercise, which promotes oxidative stress due to induction of potentially damaging biomolecules, such as proteins, lipids and DNA [44]. Recent research shows that physical inactivity also induces ROS production and may lead to muscle atrophy [45,46]. Furthermore, elevated ROS levels have been found in subjects with ageing-related sarcopenia and muscular diseases [42,47]. 

In addition, vitamin C is involved in the synthesis of carnitine and collagen [48]. While carnitine is required for the metabolism of long chain fatty acids during physical activity, collagen is a key structural component of skeletal muscles and tendons [49,50]. Animal studies have suggested that ascorbic acid deficiency in skeletal muscles caused muscle atrophy, concomitant with a high expression of muscle atrophy-related genes with a reduction in physical performance [8,51]. Interestingly, ascorbic acid supplementation restored physical performance and reduced the expression of muscle atrophy-related genes [8]. 

Limited clinical studies have determined a relationship between vitamin C deficiency and frailty. A Japanese study [9] involving 655 community dwelling older women (mean (SD) age 75.4 (4.1) years) suggested that plasma vitamin C levels positively correlate with muscle strength and physical performance measured in terms of handgrip strength, length of time standing on one leg and gait speed. Similarly, a study in the UK [52] involving 628 community dwelling patients aged 63–73 years found that a higher vitamin C intake was associated with a better physical performance measured by short chair rising time. Another European study [48], which included >13000 men and women aged 42–82 years, found that both dietary intake and plasma vitamin C levels had a positive association with fat-free mass (FFM) using bioelectrical impedance analysis.

Fruits and vegetables are rich sources of antioxidants including carotenoids, flavonoids, vitamin C and other polyphenols [53]. Research suggests that a higher intake of fruits and vegetables is protective against inflammation, cardiovascular disease and mortality [54,55,56]. It is also possible that in addition to the high oxidant stress, vitamin C deficiency also indirectly contributes to the increasing severity of frailty due to the higher prevalence of cardiovascular diseases in this population.

Although previous studies have indicated negative health outcomes with vitamin C deficiency [57,58], this was not evident in our study and clinical outcomes such as LOS, mortality and readmissions were similar across different vitamin C quintiles. Our study was, however, not powered to detect clinical outcomes. In addition, evidence to date is unconvincing that vitamin C supplementation is beneficial in improving either cardiovascular outcomes [57] or sepsis-related mortality [10,59]. However, the evidence is of low quality because of limitations such as the use of a small sample size and shorter duration of interventions. It is possible that vitamin C supplementation will be more effective for a subgroup of patients such as older people, those with a higher baseline cardiovascular risk and those with lower baseline vitamin C status. Future trials are needed avoiding the abovementioned limitations and targeting subgroup of population who are at a high risk of vitamin C deficiency to clarify beneficial effects of vitamin C supplementation.

## 5. Limitations

This study used convenient sampling for recruitment of participants, and it is possible that the sample may not be a true representative of patients who were admitted under the GEM unit. In addition, the sample is not representative of the general elderly population but of those individuals who were referred to a geriatric unit and were therefore more likely to have geriatric syndromes. The findings of this study cannot be used to suggest causality because of the cross-sectional design of the study. We were unable to quantify the intake of energy and protein of our patients in this study. There was a delay in obtaining vitamin C levels immediately following hospitalisation because micronutrient levels decline during hospitalisation due to a range of factors such as anorexia, inflammation and polypharmacy [60,61]. 

## 6. Conclusions

This study confirms that a high prevalence of hypovitaminosis C in older hospitalised patients and vitamin C deficiency is associated with a greater severity of frailty. Further studies are needed to confirm this association and to determine whether vitamin C replacement may be beneficial in prevention or reversal of frailty. 

## Figures and Tables

**Figure 1 nutrients-13-02117-f001:**
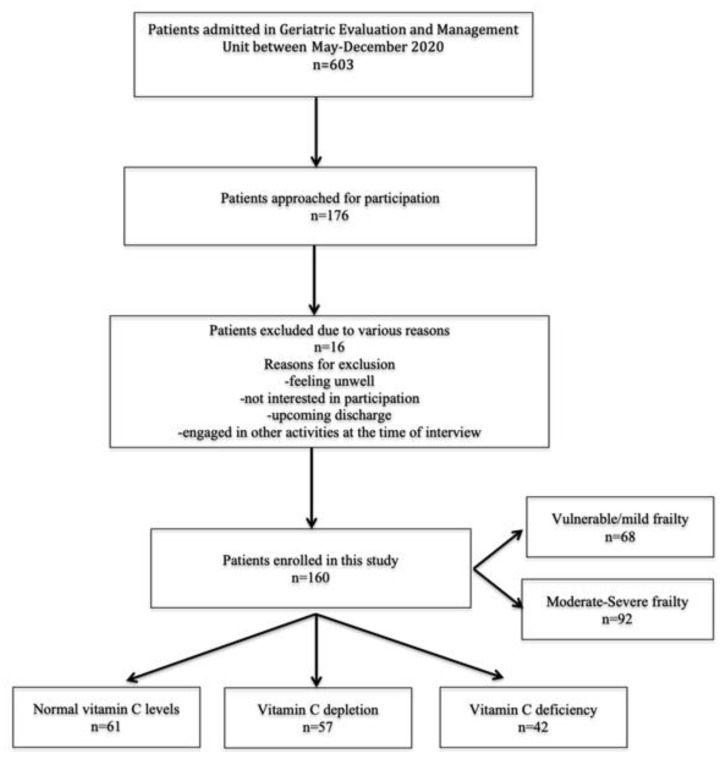
Study flow diagram.

**Figure 2 nutrients-13-02117-f002:**
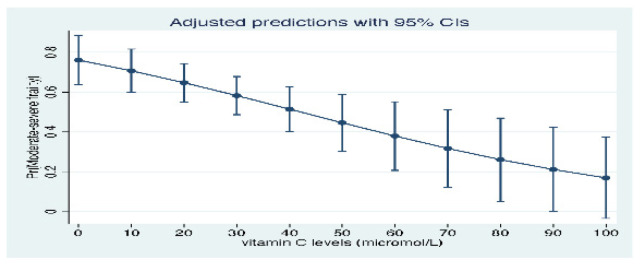
Prediction of frailty severity according to vitamin C status.

**Table 1 nutrients-13-02117-t001:** Characteristics of vulnerable/mildly frail patients compared to moderate/severely frail patients.

Variable	Non-Frail/Vulnerable/Mild Frailty	Moderate–Severe Frailty	*p* = Value
Number (%)	68 (42.5)	92 (57.5)	
Age, mean (SD)	82.8 (5.7)	85.7 (6.7)	0.004
Sex female, *n* (%)	39 (57.4)	57 (61.9)	0.557
Charlson index mean (SD)	7.8 (2.5)	8.8 (2.6)	0.001
Medications mean (SD)	6.9 (3.9)	7.9 (3.3)	0.070
Residence home alone *n* (%)	32 (47.1)	50 (54.4)	0.362
Education diploma/university *n* (%)	31 (45.6)	34 (36.9)	0.272
Income <40k/year	34 (50.8)	60 (65.2)	0.067
Medications mean (SD)	6.9 (3.9)	7.9 (3.3)	0.070
MMSE mean (SD)	25.8 (3.4)	23.8 (3.2)	<0.001
Smokers *n* (%)	41 (60.3)	51 (55.4)	0.539
GDS mean (SD)	3.5 (2.1)	5.2 (3.1)	<0.001
MUST score mean (SD)	0.86 (1.1)	0.99 (1.2)	0.511
Fruits/Vegetable intake/day mean (SD)	1.3 (0.6)	1.2 (0.6)	0.187
HARP score mean (SD)	2.4 (1.2)	3.5 (0.6)	<0.001
TUG score in seconds mean (SD)	25.3 (15.4)	40.3 (20.5)	<0.001
Vitamin C μmol/L mean (SD)	31.8 (24.4)	22.9 (21.4)	0.015
Hypovitaminosis C *n* (%)	37 (54.4)	62 (67.4)	0.095
Vitamin C deficient *n* (%)	11 (16.2)	31 (33.7)	0.013
Vitamin D nmol/L mean (SD)	62.9 (27.3)	72.4 (33.4)	0.058
Vitamin B12 pmol/L mean (SD)	442.9 (300.7)	502.5 (351.8)	0.262
Albumin g/L mean (SD)	35.6 (30.0)	31.3 (5.2)	0.180
EFS scores mean (SD)	7.7 (1.1)	11.3 (1.3)	<0.001
SPPB scores total mean (SD)	5.5 (2.8)	2.7 (2.3)	<0.001
EQ5D index mean (SD)	0.78 (0.13)	0.68 (0.16)	<0.001
LOS median (IQR)	11.5 (14)	22.7 (17)	0.004
In-hospital mortality *n* (%)	0	7 (7.6)	0.02
30-day readmissions *n* (%)	15 (22.1)	20 (21.7)	0.961

SD, Standard Deviation; MMSE, Mini Mental State Examination; GDS, Geriatric Depression Scale; MUST, Malnutrition Universal Screening Tool; HARP, Hospital Admission Risk Profile; TUG, Timed Up and Go Test; EFS, Edmonton Frail Scale; SPPB, Short Physical Performance Battery; EQ5D, European Quality of Life questionnaire; LOS, Length of Hospital Stay; IQR, Interquartile Range.

**Table 2 nutrients-13-02117-t002:** Logistic regression model showing adjusted odds ratios in moderate–severely frail patients compared to non-frail/vulnerable/mildly frail patients with normal vitamin C status as the baseline.

Variable.	aOR	95% CI	*p* Value
Vitamin C deficiency	4.30	1.33–13.86	0.015
Vitamin C depletion	1.83	0.66–5.08	0.243
Age	0.94	0.87–1.02	0.158
Sex male	0.95	0.34–2.673	0.924
Charlson index	1.03	0.85–1.27	0.715
Smokers	0.68	0.28–1.66	0.395
MUST score	1.17	0.77–1.78	0.463
MMSE score	0.89	0.78–1.02	0.104
GDS score	1.33	1.09–1.61	0.004
Fruits/Vegetable intake	0.94	0.84–1.06	0.350
HARP score	4.78	2.43–9.40	<0.001
Income <40k/year	1.67	0.63–4.44	0.302
Creatinine	0.99	0.98–1.00	0.069
Vitamin D levels	1.01	0.99–1.02	0.281
Vitamin B12 levels	1.00	0.99–1.00	0.344

aOR, adjusted odds ratio; MUST, Malnutrition Universal Screening Tool; MMSE, Mini Mental State Examination; GDS, Geriatric Depression Scale; HARP, Hospital Admission Risk Profile.

## Data Availability

The data presented in this study are available on request from the corresponding author only after permission is granted by the ethics committee.

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
