# Peer review of "Prevalence of Hypovitaminosis C and its Relationship with Frailty in Older Hospitalised Patients: A Cross-Sectional Study"

_nutrients, 2021, doi:10.3390/nu13062117_

Round 1

Reviewer 1 Report

Thank you for giving me the opportunity to review this manuscript. I think this is a interesting research topic and the manuscript is generally well-written. However, I have the serious concerns as follows.

  1. A number of potential confounders that could affect the association between vitamin C levels and frailty were not adjusted for. These include diseases, comorbidities, activities of daily living, nutritional status, intake of energy and protein, diversity in food intake. These have not been considered in any of the current studies and consequently have not been adjusted for in multivariate analyses.

  1. Considering the sample size, there are too many variables to be adjusted simultaneously in multivariate analysis. Please consult with a statistician.

  1. The calculation results are unreliable because alpha and beta errors for outcomes are not considered in the sample size calculation.

  1. Please disclose the study design.

  1. Describe how you obtained the written informed consent from the subject.

Author Response

Reviewer 1

comments and Suggestions for Authors

Thank you for giving me the opportunity to review this manuscript. I think this is a interesting research topic and the manuscript is generally well-written. However, I have the serious concerns as follows.

  1. A number of potential confounders that could affect the association between vitamin C levels and frailty were not adjusted for. These include diseases, comorbidities, activities of daily living, nutritional status, intake of energy and protein, diversity in food intake. These have not been considered in any of the current studies and consequently have not been adjusted for in multivariate analyses.

Response: We thank reviewer for this comment and agree that there are a number of variables which can be associated with frailty. We have used Charlson comorbidity index (CCI) in logistic regression model, which is based on various comorbid diseases such as myocardial infarction, congestive heart failure, diabetes, renal failure, cerebrovascular disease, peripheral vascular disease, chronic lung disease, liver disease, peptic ulcer and acquired immunodeficiency syndrome (AIDS) and is a valid and reliable method of measuring comorbidity.[1,2] This has now been explicitly stated in the Methods section page 6 paragraph 1.

“Comorbidities were assessed by use of the Charlson comorbidity index (CCI) which is a score based on various diseases such as myocardial infarction, congestive heart failure, diabetes, renal failure, cerebrovascular disease, peripheral vascular disease, chronic lung disease, liver disease, peptic ulcer and acquired immunodeficiency syndrome (AIDS) and is a valid and reliable method of measuring comorbidity.[1,2]

We have used the Malnutrition Universal Screening Tool (MUST) score in regression analysis to assess the malnutrition risk of the patients.

We were unable to quantify the intake of energy and protein for this study, however, we have now included the use of fruits and vegetables intake in the logistic regression model. We have now included this statement in the limitations section of the manuscript.

“We were unable to quantify the intake of energy and protein of the participants in this study.”

We have now also included the HARP (Hospital Admission Risk Profile) score[3] which takes into account the self-reported activities of daily living (ADL) and predicts patients at high risk of discharge to a facility, in the logistic regression analysis. This has been included in Methods section page 7 paragraph 1.

“The activities of daily living (ADL) were assessed by use of the Hospital Admission Risk Profile (HARP) score[3] which predicts patients at high risk of discharge to a facility.”

We have now included the new variables in the logistic regression model and have reanalysed the data and have modified Table 2.

  1. Considering the sample size, there are too many variables to be adjusted simultaneously in multivariate analysis. Please consult with a statistician.

Response: We have recalculated the sample size after discussion with the statistician with reference to a study by Saito et al.[4] which determined relationship between plasma vitamin C status and physical performance by measuring the handgrip strength in kg, as one of the measures of frailty. In this study, the mean (SD) handgrip strength was significantly higher among patients who were in the highest quartile for vitamin C levels compared with those who were in the lowest quartile (19.6 (4.1) vs. 17.8 (4.2), P <0.05) after adjustment for age, BMI, percent body fat, hypertension, diabetes and fruit intake. We have calculated the sample size by use of these measures, with 80% power and alpha value 0.05 the calculated sample size was 134 and with 90% power and alpha value 0.05 it was 184. These samples sizes are not much different from our previously determined sample size of 160.

  1. The calculation results are unreliable because alpha and beta errors for outcomes are not considered in the sample size calculation.

Response: Please refer to the response above for recalculation of sample size.

  1. Please disclose the study design.

Response: We have now included the study design in the title.

“Prevalence of hypovitaminosis C and its relationship with frailty in older hospitalised patients: a cross-sectional study”

  1. Describe how you obtained the written informed consent from the subject.

Response: A member of the research team approached the participants and provided participant information sheet and also provided verbal information about the research project. The participants were given sufficient time to read and discuss their participation with their carers and treating medical team and if the participants were agreeable, they were asked to sign a consent form.

This has now been clarified in the methods section page 5 paragraph 3.

“A member of the research team approached the participants and provided them with a participant information sheet in addition to verbal information about the research project. The participants were given sufficient time to read and discuss their participation with their carers as well as the treating medical team and if the participants were agreeable, they were asked to sign a consent form.”

References

  1. Frenkel, W.J.; Jongerius, E.J.; Mandjes-van Uitert, M.J.; van Munster, B.C.; de Rooij, S.E. Validation of the Charlson Comorbidity Index in acutely hospitalized elderly adults: a prospective cohort study. J. Am. Geriatr. Soc. 2014, 62, 342-346, doi:10.1111/jgs.12635.
  2. Murray, S.B.; Bates, D.W.; Ngo, L.; Ufberg, J.W.; Shapiro, N.I. Charlson Index is associated with one-year mortality in emergency department patients with suspected infection. Acad. Emerg. Med. 2006, 13, 530-536, doi:10.1197/j.aem.2005.11.084.
  3. Liu, S.K.; Montgomery, J.; Yan, Y.; Mecchella, J.N.; Bartels, S.J.; Masutani, R.; Batsis, J.A. Association Between Hospital Admission Risk Profile Score and Skilled Nursing or Acute Rehabilitation Facility Discharges in Hospitalized Older Adults. J. Am. Geriatr. Soc. 2016, 64, 2095-2100, doi:10.1111/jgs.14345.
  4. Saito, K.; Yokoyama, T.; Yoshida, H.; Kim, H.; Shimada, H.; Yoshida, Y.; Iwasa, H.; Shimizu, Y.; Kondo, Y.; Handa, S., et al. A significant relationship between plasma vitamin C concentration and physical performance among Japanese elderly women. J. Gerontol. A Biol. Sci. Med. Sci. 2012, 67, 295-301, doi:10.1093/gerona/glr174.
  5. Cook, N.R.; Albert, C.M.; Gaziano, J.M.; Zaharris, E.; MacFadyen, J.; Danielson, E.; Buring, J.E.; Manson, J.E. A Randomized Factorial Trial of Vitamins C and E and Beta Carotene in the Secondary Prevention of Cardiovascular Events in Women: Results From the Women's Antioxidant Cardiovascular Study. Arch. Intern. Med. 2007, 167, 1610-1618, doi:10.1001/archinte.167.15.1610.
  6. Loria, C.M.; Klag, M.J.; Caulfield, L.E.; Whelton, P.K. Vitamin C status and mortality in US adults. Am. J. Clin. Nutr. 2000, 72, 139-145, doi:10.1093/ajcn/72.1.139.
  7. Ashor, A.W.; Brown, R.; Keenan, P.D.; Willis, N.D.; Siervo, M.; Mathers, J.C. Limited evidence for a beneficial effect of vitamin C supplementation on biomarkers of cardiovascular diseases: an umbrella review of systematic reviews and meta-analyses. Nutr. Res. 2019, 61, 1-12, doi:10.1016/j.nutres.2018.08.005.
  8. Scholz, S.S.; Borgstedt, R.; Ebeling, N.; Menzel, L.C.; Jansen, G.; Rehberg, S. Mortality in septic patients treated with vitamin C: a systematic meta-analysis. Crit. Care 2021, 25, 17, doi:10.1186/s13054-020-03438-9.

Reviewer 2 Report

The manuscript by Sharma and colleagues presents the results of a cross-sectional observational study of individuals admitted to the geriatric unit of Flinders Medical Center, Adelaide, South Australia. 176 individuals >75 years of age were recruited and 160 patients participated in the study. Participants were classified on frailty through the Edmonton Frail Scale and the score was correlated with various characteristics, showing significant associations with age, Charlson index, MMSE, GDS, EFS, SPPB, TUG and EQ5D. Moreover, plasma vitamin C levels were assessed and classified into 3 groups: repleted, depleted or deficient. There was an inverse correlation between participants' frailty and vitamin C levels. This correlation remained significant using the logistic regression model which showed odds ratio adjusted for age, sex, vit D, vit B12, MMSE, and other possible confounders.

In addition to known nutritional and lifestyle risk factors on frailty in older people, vitamin C levels may suggest a micronutrient deficiency that needs attention.

The work is smooth and well written. The supporting literature is adequate. The results are well exposed and the design of the study is clear and exhaustive.

I just want to suggest some minor aspects that deserve attention and could be adequately clarified:

- In the classification of EFS scores, the authors specify that individuals with a score of 0-5 are considered “not frail”. However, the authors indicate that the participants showed an EFS score with a range of 5-16 and classified the patients into two categories (vulnerable/mildly frail or moderate-severely frail). If the range is correct, I assume there will be participants who scored 5 and therefore should not have been included in the two groupings. This aspect confuses the reader. Moreover, the authors stated that the participants were divided into only two categories of fragility (with scores <10 or ≥10) and that therefore the presence of non-fragile individuals is excluded. In any case, the two categories would be more correctly described as between 6 and 9 or ≥10.

- In the flow chart and in the text, it emerges that 176 out of 603 participants were approached admitted to the unit in the reference period. How were they selected? It is an important aspect to understand if the subset of 176 participants was representative.

- Perhaps it depends on my unfamiliarity with the type of analysis carried out in Figure 2, but is it possible to specify whether there is a significant linear correlation between vitamin C levels and the fragility of patients?

- The lack of a nutritional investigation is very limiting because it does not allow us to hypothesize whether the correlation with vitamin C depends on a reduced intake or lower absorption and metabolization. This aspect may be suggested for future works

- The cross-sectional observational nature of the study, which does not allow to highlight causal relationships, should also be highlighted among the limits

- Furthermore, among the limits, it should be specified that the sample is not representative of the general elderly population but of those individuals who have referred to a geriatric unit and therefore with potential geriatric disorders.

- In defining the role of exogenous antioxidants, I would be more careful to suggest possible intake through concentrated supplements that do not currently appear to show solid evidence of long-term benefits (DOI: 10.1001 / jama.297.8.842).

Author Response

The manuscript by Sharma and colleagues presents the results of a cross-sectional observational study of individuals admitted to the geriatric unit of Flinders Medical Center, Adelaide, South Australia. 176 individuals >75 years of age were recruited and 160 patients participated in the study. Participants were classified on frailty through the Edmonton Frail Scale and the score was correlated with various characteristics, showing significant associations with age, Charlson index, MMSE, GDS, EFS, SPPB, TUG and EQ5D. Moreover, plasma vitamin C levels were assessed and classified into 3 groups: repleted, depleted or deficient. There was an inverse correlation between participants' frailty and vitamin C levels. This correlation remained significant using the logistic regression model which showed odds ratio adjusted for age, sex, vit D, vit B12, MMSE, and other possible confounders.

In addition to known nutritional and lifestyle risk factors on frailty in older people, vitamin C levels may suggest a micronutrient deficiency that needs attention.

The work is smooth and well written. The supporting literature is adequate. The results are well exposed and the design of the study is clear and exhaustive.

I just want to suggest some minor aspects that deserve attention and could be adequately clarified:

- In the classification of EFS scores, the authors specify that individuals with a score of 0-5 are considered “not frail”. However, the authors indicate that the participants showed an EFS score with a range of 5-16 and classified the patients into two categories (vulnerable/mildly frail or moderate-severely frail). If the range is correct, I assume there will be participants who scored 5 and therefore should not have been included in the two groupings. This aspect confuses the reader. Moreover, the authors stated that the participants were divided into only two categories of fragility (with scores <10 or ≥10) and that therefore the presence of non-fragile individuals is excluded. In any case, the two categories would be more correctly described as between 6 and 9 or ≥10.

Response: We thank reviewer for this correction and for more clarity have now changed the heading EFS score <10 as non-frail/vulnerable/mild frailty vs. ≥10 as moderate/severe frailty in the abstract, main text, tables and Figure 1.

- In the flow chart and in the text, it emerges that 176 out of 603 participants were approached admitted to the unit in the reference period. How were they selected? It is an important aspect to understand if the subset of 176 participants was representative.

Response: We used convenience sampling for recruitment and only 176 out of 603 participants were approached for participation. The recruitment was limited by the resources and the capacity of a single registrar to thoroughly process recruits to this study over its seven months operation.  We understand that this a limitation of this study and have now included this statement in the limitations section.

“This study used convenient sampling for recruitment of participants and it is possible that the sample may not be a true representative of patients who were admitted under the GEM unit.”

- Perhaps it depends on my unfamiliarity with the type of analysis carried out in Figure 2, but is it possible to specify whether there is a significant linear correlation between vitamin C levels and the fragility of patients?

Response: There was a weak negative correlation between vitamin C levels and EFS score (correlation coefficient = - 0.14, P = 0.081) and we have not plotted the graph (Figure below) because it does not look impressive.

- The lack of a nutritional investigation is very limiting because it does not allow us to hypothesize whether the correlation with vitamin C depends on a reduced intake or lower absorption and metabolization. This aspect may be suggested for future works

Response: We agree that that we were not able to quantify the protein and energy intake of the participants in this study and have included this as a limitation. However, we have now included the intake of fruits/vegetables as an additional variable and have reanalysed the data.

“We were unable to quantify the intake of energy and protein of our patients in this study.”

- The cross-sectional observational nature of the study, which does not allow to highlight causal relationships, should also be highlighted among the limits

Response: We have included a statement in the limitations section.

“The findings of this study cannot be used to suggest causality because of the cross-sectional design of the study.”

- Furthermore, among the limits, it should be specified that the sample is not representative of the general elderly population but of those individuals who have referred to a geriatric unit and therefore with potential geriatric disorders.

Response: We have included a statement in the limitations section as advised by the reviewer.

“In addition, the sample is not representative of the general elderly population but of those individuals, who were referred to a geriatric unit, and, were therefore more likely to have geriatric syndromes.”

- In defining the role of exogenous antioxidants, I would be more careful to suggest possible intake through concentrated supplements that do not currently appear to show solid evidence of long-term benefits (DOI: 10.1001 / jama.297.8.842).

Response: We thank reviewer for this update and have now included a paragraph in discussion section

“Although previous studies have indicated negative health outcomes with vitamin C deficiency [56,57], evidence till date is unconvincing that vitamin C supplementation is beneficial in improving either cardiovascular outcomes [56] or sepsis related mortality [10,58]. However, the evidence is of low quality because of limitations such as use of a small sample size and shorter duration of interventions. It is possible that vitamin C supplementation will be more effective for a subgroup of patients such as older people, those with a higher baseline cardiovascular risk and those with lower baseline vitamin C status. Future trials are needed avoiding the above mentioned limitations and targeting subgroup of population who are at a high risk of vitamin C deficiency to clarify beneficial effects of vitamin C supplementation.”

Reviewer 3 Report

I have reviewed the manuscript “Prevalence of hypovitaminosis C and its relationship with frailty in older hospitalised patients: an observational study”, which aims to study the prevalence of vitamin C deficiency and its association with frailty severity in patients admitted under a geriatric unit. The study topic is relevant, the study question is interesting, the manuscript is straightforward and well written, although the discussion rather dry to me. I do have a couple of major concerns that refrain my enthusiasm about the results and need revision in order to provide the readership with a trustful interpretation.

-I am very interested in looking more carefully at the subgroup of patients on the upper end of plasma vitamin C levels. It seems to me that all patients with vitamin C levels higher than 28 umol/L were treated as one uniform group. I do not think this is correct because it does not really distinguish between actually normal and supraphysiological plasma levels of vitamin C. Recent studies have shown that the latter might actually associate with negative outcomes. In the current version of the manuscript it is not possible to discern on this potential issue.

-also in relation to the previous comment, according to figure 2, it seems that the beneficial effect of vitamin C is not apparent after reaching certain plasma levels. I think this issue deserves further discussion, analyses taking into account recent literature on the topic, and original proposals for future directions with careful consideration of a potentially limiting effect of vitamin C-targeted therapies.

-a very important confounder not accounted for in the study is socioeconomic status. I highly doubt the results of this study based on the logistic regression analyses are trustworthy without removing such effect in multivariable-adjusted analyses.

-please also add MMSE to the multivariable adjusted logistic regression analyses

-I would also be interested in evaluating a potential effect-modification by vitamins B12 and D.

-please disclose missing data and add mentioning on how missing data was handled

Other comments:

-please clearly indicate in the title that this is a cross-sectional study

-I think the abstract would benefit with incorporating a scientific rationale to hold the plea for studying the association between vitamin C and frailty. Although there is a proposal in the introduction section, I think it is now missing in the abstract.

-I would also –at least very briefly– mention whether previous studies have investigated this specific question. Please also add that to the main text.

-I think it would also be beneficial to mention clearly in the abstract that study participants were 75 years old or older.

-please provide in the methods section the starting and finishing date of the study. After looking this up for a while, I came to realize that it would be mentioned after in the results section, which I found rather odd as the recruitment period is not really a result of the study.

-at the moment it is not fully clear to me why only 176 patients were approached for participation among 603 patients admitted to the GEM unit. Please clearly indicate whether inclusion and exclusion criteria were the only factors determining which patients were approached for participation. Please also clearly indicate whether the patients were consecutively approached for participation during the recruitment period.

-how come that the age range goes from 73 years-old, while the age recruiting criteria was ≥ 75 years-old (materials and methods)

Author Response

I have reviewed the manuscript “Prevalence of hypovitaminosis C and its relationship with frailty in older hospitalised patients: an observational study”, which aims to study the prevalence of vitamin C deficiency and its association with frailty severity in patients admitted under a geriatric unit. The study topic is relevant, the study question is interesting, the manuscript is straightforward and well written, although the discussion rather dry to me. I do have a couple of major concerns that refrain my enthusiasm about the results and need revision in order to provide the readership with a trustful interpretation.

-I am very interested in looking more carefully at the subgroup of patients on the upper end of plasma vitamin C levels. It seems to me that all patients with vitamin C levels higher than 28 umol/L were treated as one uniform group. I do not think this is correct because it does not really distinguish between actually normal and supraphysiological plasma levels of vitamin C. Recent studies have shown that the latter might actually associate with negative outcomes. In the current version of the manuscript it is not possible to discern on this potential issue.

Response: We thank reviewer for this comment and divided vitamin C status into quintiles and determined clinical outcomes in our patients: length of hospital stay, 30-day mortality and 30-day readmission rates and found that these outcomes were not significantly different between 5 groups (P value >0.05). Although, we acknowledge that there were only a few (7) deaths and our sample was not powered to look into clinical outcomes. Another study is needed to further clarify this issue.

-also in relation to the previous comment, according to figure 2, it seems that the beneficial effect of vitamin C is not apparent after reaching certain plasma levels. I think this issue deserves further discussion, analyses taking into account recent literature on the topic, and original proposals for future directions with careful consideration of a potentially limiting effect of vitamin C-targeted therapies.

Response: We have now included this as a separate paragraph in discussion pages 18-19 discussion section

“Although previous studies have indicated negative health outcomes with vitamin C deficiency[5,6], evidence till date is unconvincing that vitamin C supplementation is beneficial in improving either cardiovascular outcomes [5] or sepsis related mortality [7,8]. However, the evidence is of low quality because of limitations such as use of a small sample size and shorter duration of interventions. It is possible that vitamin C supplementation will be more effective for a subgroup of patients such as older people, those with a higher baseline cardiovascular risk and those with lower baseline vitamin C status. Future trials are needed avoiding the above mentioned limitations and targeting subgroup of population who are at a high risk of vitamin C deficiency to clarify beneficial effects of vitamin C supplementation.”

-a very important confounder not accounted for in the study is socioeconomic status. I highly doubt the results of this study based on the logistic regression analyses are trustworthy without removing such effect in multivariable-adjusted analyses.

Response: We agree with the reviewer and have now included socioeconomic status in the multivariable analyses in Table 2.

-please also add MMSE to the multivariable adjusted logistic regression analyses

Response: MMSE is now included as per suggestion and data has been reanalysed.

-I would also be interested in evaluating a potential effect-modification by vitamins B12 and D.

Response: We reanalysed the data by including interaction terms between vitamin B12 and vitamin D and found no significant effect modification.

-please disclose missing data and add mentioning on how missing data was handled

Response: There was no missing data and vitamin C levels were available for all the participants. This statement has now been included in the Results section page 9.

“There was no missing data and vitamin C results were available for all the participants.”

Other comments:

-please clearly indicate in the title that this is a cross-sectional study

Response: We have now included this in the heading.

“Prevalence of hypovitaminosis C and its relationship with frailty in older hospitalised patients: a cross-sectional study”

-I think the abstract would benefit with incorporating a scientific rationale to hold the plea for studying the association between vitamin C and frailty. Although there is a proposal in the introduction section, I think it is now missing in the abstract.

Response: We have now included this in the abstract as suggested by the reviewer.

“Frailty is common in older hospitalised patients and may be associated with micronutrient malnutrition.”

-I would also –at least very briefly– mention whether previous studies have investigated this specific question. Please also add that to the main text.

Response: We have now included this in the abstract and introduction section.

Abstract

“Only limited studies have explored the relationship between frailty and vitamin C deficiency.”

Introduction

“Till date, only limited studies[4,8]  have explored relationship between frailty and vitamin C deficiency.”

-I think it would also be beneficial to mention clearly in the abstract that study participants were 75 years old or older.

Response: This has now been clearly stated as advised by the reviewer.

“This study investigated the prevalence of vitamin C deficiency and its association with frailty severity in patients ≥75 years admitted under a geriatric unit.”

-please provide in the methods section the starting and finishing date of the study. After looking this up for a while, I came to realize that it would be mentioned after in the results section, which I found rather odd as the recruitment period is not really a result of the study.

Response: The starting and finishing dates are included in the methods section as advised by the reviewer.

“Patients ≥ 75 years who were admitted to the GEM unit of Flinders Medical Centre between May-December 2020 were recruited by convenience sampling in this research.”

-at the moment it is not fully clear to me why only 176 patients were approached for participation among 603 patients admitted to the GEM unit. Please clearly indicate whether inclusion and exclusion criteria were the only factors determining which patients were approached for participation. Please also clearly indicate whether the patients were consecutively approached for participation during the recruitment period.

Response: This study recruited patients by convenience sampling and only 176 patients were approached for participation, of whom, 16 patients were excluded based on inclusion and exclusion criteria and 160 patients were finally enrolled.

This has been now explicitly stated in the results section

Six hundred and three patients were admitted under the GEM unit between May-December 2020, of whom, 176 patients were approached by convenience sampling for participation and 160 patients were recruited for this study while 16 patients were excluded because of various reasons (Figure 1).

-how come that the age range goes from 73 years-old, while the age recruiting criteria was ≥ 75 years-old (materials and methods)

Response: This was an error and has now been corrected.

Round 2

Reviewer 1 Report

Thank you for giving me the opportunity to review the revised manuscript. As I pointed out in the previous article, there were too many adjustment variables to be fed into the logistic regression analysis at once. The logistic regression model may require a sample size such that the lesser number of binary outcomes is at least "number of exposure factors x 10". Moreover, it is not desirable to include independent variables that were highly correlated with other independent variables. For example, vitamin C deficiency and vitamin C depletion, Charlson Index and CRP levels, and MUST score and albumin levels would be highly correlated and should not be included at the same time.

Author Response

Thank you for giving me the opportunity to review the revised manuscript. As I pointed out in the previous article, there were too many adjustment variables to be fed into the logistic regression analysis at once. The logistic regression model may require a sample size such that the lesser number of binary outcomes is at least "number of exposure factors x 10". Moreover, it is not desirable to include independent variables that were highly correlated with other independent variables. For example, vitamin C deficiency and vitamin C depletion, Charlson Index and CRP levels, and MUST score and albumin levels would be highly correlated and should not be included at the same time.

Response

We thank reviewer for constructive feedback. We agree with the reviewer that owing to the small sample size of our study the association between exposure variables and outcome could have been overestimated by employing logistic regression model. We have now applied corrective measures as suggested by the reviewer by removing variables which could be correlated. We have removed C-RP which can be correlated with Charlson index and albumin levels which can correlate with the MUST score. In addition, we have now determined a bias corrected estimate using the jackknife resampling method as suggested by Nemes et al [1]. This has now been included as sensitivity analysis.

Please refer to statistics section page 8

“We performed sensitivity analysis and determined the bias corrected estimates using the jackknife resampling method as suggested by Nemes et al [1].”

“Senstivity analysis using the jackknife resampling method confirmed these estimates (aOR 4.30, SE 3.01, 95% CI 1.07 – 17.12, P = 0.039).”

Reference

  1. Nemes, S.; Jonasson, J.M.; Genell, A.; Steineck, G. Bias in odds ratios by logistic regression modelling and sample size. BMC Med. Res. Methodol. 2009, 9, 56, doi:10.1186/1471-2288-9-56.

Reviewer 2 Report

The authors improved the manuscript in accordance with the revision requests

Author Response

We thank reviewer for constructive feedback.

Reviewer 3 Report

Thank you for taking my comments into account. 

Yet, I think it is important that all the comments are taken into account not just to reply to me, but also to actually review the manuscript. This was indeed the case for most of my comments, except a few. 

Author Response

Thank you for taking my comments into account. 

Yet, I think it is important that all the comments are taken into account not just to reply to me, but also to actually review the manuscript. This was indeed the case for most of my comments, except a few. 

Comment We thank reviewer for the constructive feedback and have now included all the suggestions in the manuscript.

I have reviewed the manuscript “Prevalence of hypovitaminosis C and its relationship with frailty in older hospitalised patients: an observational study”, which aims to study the prevalence of vitamin C deficiency and its association with frailty severity in patients admitted under a geriatric unit. The study topic is relevant, the study question is interesting, the manuscript is straightforward and well written, although the discussion rather dry to me. I do have a couple of major concerns that refrain my enthusiasm about the results and need revision in order to provide the readership with a trustful interpretation.

-I am very interested in looking more carefully at the subgroup of patients on the upper end of plasma vitamin C levels. It seems to me that all patients with vitamin C levels higher than 28 umol/L were treated as one uniform group. I do not think this is correct because it does not really distinguish between actually normal and supraphysiological plasma levels of vitamin C. Recent studies have shown that the latter might actually associate with negative outcomes. In the current version of the manuscript it is not possible to discern on this potential issue.

Comment: We thank reviewer for this comment and divided vitamin C status into quintiles and determined clinical outcomes in our patients: length of hospital stay (LOS), inhospital mortality and 30-day readmission rates and found that these outcomes were not significantly different between 5 groups (P value >0.05). Although, we acknowledge that there were only a few (7) deaths and our sample was not powered to look into clinical outcomes. Another study is needed to further clarify this issue.

We have now included this in methods section page 7, Statistics section (page 8), Table 1 depicting LOS, inhospital mortality and 30-day readmissions

“Vitamin C levels were also divided into quintiles and we compared clinical outcomes such as length of hospital stay (LOS), mortality and readmissions within 30 days of hospital discharge in different quintiles.”

Statistics section (page 8)

“Kruskal Wallis H test was used to compare LOS in different quintiles of vitamin C.”

Results page 9

“Both LOS and in hospital mortality was significantly higher among patients who were moderately to severely frail in comparison with non-frail/vulnerable/mildly frail group, however, there was no difference in 30-day readmissions between the two groups.”

Results page 13

“Clinical outcomes such as LOS, inhospital mortality and 30-day readmissions were not significantly different according to different vitamin C quintiles (P > 0.05).”

-also in relation to the previous comment, according to figure 2, it seems that the beneficial effect of vitamin C is not apparent after reaching certain plasma levels. I think this issue deserves further discussion, analyses taking into account recent literature on the topic, and original proposals for future directions with careful consideration of a potentially limiting effect of vitamin C-targeted therapies.

Comment-We have now included this as a separate paragraph in discussion pages 18-19 discussion section

“Although previous studies have indicated negative health outcomes with vitamin C deficiency[5,6], this was not evident in our study as clinical outcomes such as LOS, mortality and readmissions were similar across different vitamin C quintiles. Our study was, however, not powered to detect clinical outcomes. In addition,  evidence till date is unconvincing that vitamin C supplementation is beneficial in improving either cardiovascular outcomes [5] or sepsis related mortality [7,8]. However, the evidence is of low quality because of limitations such as use of a small sample size and shorter duration of interventions. It is possible that vitamin C supplementation will be more effective for a subgroup of patients such as older people, those with a higher baseline cardiovascular risk and those with lower baseline vitamin C status. Future trials are needed avoiding the above mentioned limitations and targeting subgroup of population who are at a high risk of vitamin C deficiency to clarify beneficial effects of vitamin C supplementation.”

-a very important confounder not accounted for in the study is socioeconomic status. I highly doubt the results of this study based on the logistic regression analyses are trustworthy without removing such effect in multivariable-adjusted analyses.

Comment: We agree with the reviewer and have now included socioeconomic status in the multivariable analyses in Table 2.

Please refer to Table 2 and a new variable “Income <40k/year” has been included.

-please also add MMSE to the multivariable adjusted logistic regression analyses

Comment: MMSE is now included as per suggestion and data has been reanalysed.

Please refer to Table 2 MMSE has been included in logistic regression analysis

-I would also be interested in evaluating a potential effect-modification by vitamins B12 and D.

Comment: We reanalysed the data by including interaction terms between vitamin B12 and vitamin D and found no significant effect modification. This has been included in the results section. This has been included in the statistics and results section on pages 8 and 13.

“We determined  any effect modification by use of interaction terms with vitamin D and vitamin B12 in the model.”

“There was no effect modification with the use of interaction terms with vitamin B12 and vitamin D.”

-please disclose missing data and add mentioning on how missing data was handled

Comment: There was no missing data and vitamin C levels were available for all the participants. This statement has now been included in the Results section page 9.

“There was no missing data and vitamin C results were available for all the participants.”

Other comments:

-please clearly indicate in the title that this is a cross-sectional study

Comment: We have now included this in the heading.

“Prevalence of hypovitaminosis C and its relationship with frailty in older hospitalised patients: a cross-sectional study”

-I think the abstract would benefit with incorporating a scientific rationale to hold the plea for studying the association between vitamin C and frailty. Although there is a proposal in the introduction section, I think it is now missing in the abstract.

Comment: We have now included this in the abstract as suggested by the reviewer.

“Frailty is common in older hospitalised patients and may be associated with micronutrient malnutrition.”

-I would also –at least very briefly– mention whether previous studies have investigated this specific question. Please also add that to the main text.

Comment: We have now included this in the abstract and introduction section.

Abstract

“Only limited studies have explored the relationship between frailty and vitamin C deficiency.”

Introduction

“Till date, only limited studies[4,8]  have explored relationship between frailty and vitamin C deficiency.”

-I think it would also be beneficial to mention clearly in the abstract that study participants were 75 years old or older.

Comment: This has now been clearly stated as advised by the reviewer.

“This study investigated the prevalence of vitamin C deficiency and its association with frailty severity in patients ≥75 years admitted under a geriatric unit.”

-please provide in the methods section the starting and finishing date of the study. After looking this up for a while, I came to realize that it would be mentioned after in the results section, which I found rather odd as the recruitment period is not really a result of the study.

Comment: The starting and finishing dates are included in the methods section as advised by the reviewer.

“Patients ≥ 75 years who were admitted to the GEM unit of Flinders Medical Centre between May-December 2020 were recruited by convenience sampling in this research.”

-at the moment it is not fully clear to me why only 176 patients were approached for participation among 603 patients admitted to the GEM unit. Please clearly indicate whether inclusion and exclusion criteria were the only factors determining which patients were approached for participation. Please also clearly indicate whether the patients were consecutively approached for participation during the recruitment period.

Comment: This study recruited patients by convenience sampling and only 176 patients were approached for participation, of whom, 16 patients were excluded based on inclusion and exclusion criteria and 160 patients were finally enrolled.

This has been now explicitly stated in the results section (page 9).

“Six hundred and three patients were admitted under the GEM unit between May-December 2020, of whom, 176 patients were approached by convenience sampling for participation and 160 patients were recruited for this study while 16 patients were excluded because of various reasons (Figure 1).”

-how come that the age range goes from 73 years-old, while the age recruiting criteria was ≥ 75 years-old (materials and methods)

Comment: This was an error and has now been corrected.

References

  1. Frenkel, W.J.; Jongerius, E.J.; Mandjes-van Uitert, M.J.; van Munster, B.C.; de Rooij, S.E. Validation of the Charlson Comorbidity Index in acutely hospitalized elderly adults: a prospective cohort study. J. Am. Geriatr. Soc. 2014, 62, 342-346, doi:10.1111/jgs.12635.
  2. Murray, S.B.; Bates, D.W.; Ngo, L.; Ufberg, J.W.; Shapiro, N.I. Charlson Index is associated with one-year mortality in emergency department patients with suspected infection. Acad. Emerg. Med. 2006, 13, 530-536, doi:10.1197/j.aem.2005.11.084.
  3. Liu, S.K.; Montgomery, J.; Yan, Y.; Mecchella, J.N.; Bartels, S.J.; Masutani, R.; Batsis, J.A. Association Between Hospital Admission Risk Profile Score and Skilled Nursing or Acute Rehabilitation Facility Discharges in Hospitalized Older Adults. J. Am. Geriatr. Soc. 2016, 64, 2095-2100, doi:10.1111/jgs.14345.
  4. Saito, K.; Yokoyama, T.; Yoshida, H.; Kim, H.; Shimada, H.; Yoshida, Y.; Iwasa, H.; Shimizu, Y.; Kondo, Y.; Handa, S., et al. A significant relationship between plasma vitamin C concentration and physical performance among Japanese elderly women. J. Gerontol. A Biol. Sci. Med. Sci. 2012, 67, 295-301, doi:10.1093/gerona/glr174.
  5. Cook, N.R.; Albert, C.M.; Gaziano, J.M.; Zaharris, E.; MacFadyen, J.; Danielson, E.; Buring, J.E.; Manson, J.E. A Randomized Factorial Trial of Vitamins C and E and Beta Carotene in the Secondary Prevention of Cardiovascular Events in Women: Results From the Women's Antioxidant Cardiovascular Study. Arch. Intern. Med. 2007, 167, 1610-1618, doi:10.1001/archinte.167.15.1610.
  6. Loria, C.M.; Klag, M.J.; Caulfield, L.E.; Whelton, P.K. Vitamin C status and mortality in US adults. Am. J. Clin. Nutr. 2000, 72, 139-145, doi:10.1093/ajcn/72.1.139.
  7. Ashor, A.W.; Brown, R.; Keenan, P.D.; Willis, N.D.; Siervo, M.; Mathers, J.C. Limited evidence for a beneficial effect of vitamin C supplementation on biomarkers of cardiovascular diseases: an umbrella review of systematic reviews and meta-analyses. Nutr. Res. 2019, 61, 1-12, doi:10.1016/j.nutres.2018.08.005.
  8. Scholz, S.S.; Borgstedt, R.; Ebeling, N.; Menzel, L.C.; Jansen, G.; Rehberg, S. Mortality in septic patients treated with vitamin C: a systematic meta-analysis. Crit. Care 2021, 25, 17, doi:10.1186/s13054-020-03438-9.